# Synthesis of Porous Material from Coal Gasification Fine Slag Residual Carbon and Its Application in Removal of Methylene Blue

**DOI:** 10.3390/molecules26206116

**Published:** 2021-10-10

**Authors:** Yixin Zhang, Rumeng Wang, Guofeng Qiu, Wenke Jia, Yang Guo, Fanhui Guo, Jianjun Wu

**Affiliations:** 1National Engineering Research Center of Coal Preparation and Purification, China University of Mining and Technology, No. 1 Daxue Road, Xuzhou 221116, China; yixinzhang@cumt.edu.cn; 2Shandong Xuanyuan Scientific Engineering and Industrial Technology Research Institute Co., Ltd., Longgu, Juye, Heze 274918, China; 3School of Chemical Engineering and Technology, China University of Mining and Technology, No. 1 Daxue Road, Xuzhou 221116, China; rumengwang@cumt.edu.cn (R.W.); ts20040045a31@cumt.edu.cn (G.Q.); jwk0812@163.com (W.J.); cumt-guoyang@cumt.edu.cn (Y.G.)

**Keywords:** coal gasification slag, residual carbon activation, methylene blue adsorption, kinetics, isotherms

## Abstract

A large amount of coal gasification slag is produced every year in China. However, most of the current disposal is into landfills, which causes serious harm to the environment. In this research, coal gasification fine slag residual carbon porous material (GFSA) was prepared using gasification fine slag foam flotation obtained carbon residue (GFSF) as raw material and an adsorbent to carry out an adsorption test on waste liquid containing methylene blue (MB). The effects of activation parameters (GFSF/KOH ratio mass ratio, activation temperature, and activation time) on the cation exchange capacity (CEC) of GFSA were investigated. The total specific surface area and pore volume of GSFA with the highest CEC were 574.02 m^2^/g and 0.467 cm^3^/g, respectively. The degree of pore formation had an important effect on CEC. The maximum adsorption capacity of GFSA on MB was 19.18 mg/g in the MB adsorption test. The effects of pH, adsorption time, amount of adsorbent, and initial MB concentration on adsorption efficiency were studied. Langmuir isotherm and quasi second-order kinetic model have a good fitting effect on the adsorption isotherm and kinetic model of MB.

## 1. Introduction

Coal is one of the most economical and safe primary energies that can be used cleanly and efficiently. It is widely used in many industries such as electric power, iron and steel, metallurgy, chemical, and building materials. Coal accounts for more than 50% of China′s primary energy production and consumption structure, and China is the largest consumer of coal [1,2]. With the change in the world′s climate and environment, the clean and efficient use of coal is particularly important.

Coal gasification technology is an important part of the clean and efficient utilization of coal resources. The residue produced in the process of coal gasification includes fine slag (GFS) and coarse slag (GCS). GCS is produced in the slag discharge mouth of the gasifier; GFS is mainly produced in the dust removal device of syngas [3,4]. Until now, the main treatment methods for gasification slag are storage and disposal in a landfill, which cause serious environmental pollution and waste of land resources. Therefore, it is urgent to develop economic, environmental, and efficient treatment methods for gasification slag [5,6,7,8].

Researchers have studied the physical and chemical properties of GFS [4,9,10,11,12]. GFS is composed of mineral-rich particles and residual carbon [13]. The content of residual carbon in GFS can reach more than 30%. The mineral-rich particles in GFS mainly consist of crystalline minerals (silicates, aluminosilicates, and Ca–Fe oxides) and vitreous components (Ca–Fe–aluminosilicate glass) [9]. At present, most GFS is used by blending. The utilization of building materials and the mixing of circulating fluidized bed boilers are the main ways coal gasification slag is used [14]. The residual carbon in the GFS has been carbonized to a certain extent in the high-temperature gasifier. Wagner, et al. [15] discovered that the residual carbon in gasification slag has a high specific surface area and micropore area, which could potentially be used as activated carbon or precursors for premium carbon products. The use of residual carbon in coal gasification slag to prepare porous materials not only provides a new possibility for the recycling of GFS but also eliminates a certain environmental pressure. There has been some research on the preparation of activated carbon from gasification slag [16,17,18,19]. Miao, et al. [17] used GFS as a raw material to prepare activated carbon through acid treatment and KOH activation, which was used for CO_2_ capture. Xu and Chai [18] prepared coal gasification slag-based activated carbon loaded with Fe^3+^ by an impregnation method, and achieved good results in the degradation of methyl orange in dye wastewater. However, although the materials obtained by these processes can obtain better properties, they cannot meet the requirements of large-scale industrialization due to their complicated operation and high cost.

With the rapid development of the printing and dyeing industry, a large amount of dye wastewater has seriously endangered human health and the environment [20]. Methylene blue (MB) is a cationic dye used for dyeing cotton, hemp, silk, paper, and so on. MB has some harmful effects on the body, including burning of the eyes, which can cause permanent damage, increased heart rate, nausea, vomiting, and shock [21,22,23]. At present, the treatment technology for dye wastewater includes physical (adsorption and membrane filtration), chemical (coagulation/flocculation and chemical oxidation), and biological methods [24,25]. Among them, adsorption is considered to be one of the simplest, most effective, and least costly technologies [26]. The adsorption process is simple, the equipment requirements are low, and solid waste can be used as raw material synthesis adsorbent, such as fly ash to synthesize zeolite, and activated carbon from coal gasification slag [17,27]. The cation exchange capacity (CEC) can be determined for the evaluation of the performance of the adsorbent. CEC depends on the number of adsorption sites of the material, which can be used as one of the indicators to evaluate the performance of adsorption materials [28,29,30].

In this study, using gasification fine slag foam flotation obtained carbon residue (GFSF) as raw material, the coal gasification fine slag residual carbon porous materials (GFSA) were prepared by alkali fusion–hydrothermal reaction, and the influence of some parameters (GFSF/KOH ratio, activation temperature and activation time) of the alkali fusion process was investigated. SEM and BET characterized the porosity of the material. The optimal product was used to remove MB from simulated wastewater. The effects of pH, adsorption time, amount of adsorbent, and initial MB concentration on adsorption efficiency were studied and the adsorption behavior is discussed.

## 2. Experiment

### 2.1. Raw Material

The GFS sample was obtained from an entrained flow bed gasifier in Shenhua Ningxia Coal Industry Group Co., Ltd., Yinchuan, China. The raw material was dried at 105 °C for 6 h to remove moisture and stored in a sealed bag at room temperature.

The foam flotation test was used to pretreat GFS. The froth floatation test was carried out by an XFD-type floatation machine (Ganzhou, China) having a capacity of 1 L and the slurry concentration was maintained at 100 g/L. The dried GFS was mixed with distilled water and poured into the flotation tank. The grout was first stirred at a speed of 1800 r/min for 2 min, and then a certain amount of collector (7 kg/t) and frother (14 kg/t) were added successively. After stirring evenly, the air valve was opened to keep the gas flow rate at 0.2 m^3^/h, and the bubbles were scraped. After repeating three times, the residual carbon and tail ash were obtained [31]. The loss on ignition of the residual carbon was 69.54%. The gasification fine slag foam flotation obtained carbon residue (GFSF) was used in subsequent experiments. The separated tail ash had a higher ash purity and could be used to prepare zeolite, as detailed in our previous study [30]. The loss on ignition of GFS and GFSF was 26.12% and 69.54%.

Potassium hydroxide (KOH), sodium hydroxide (NaOH), and hydrochloric acid (HCl) were AR-grade, purchased from Aladdin Co., Ltd. (Shanghai, China). Methylene blue trihydrate (MB) was AR-grade from Tianjin Kemiou Chemical Reagent Co., Ltd. (Tianjin, China). Collector (W501) and frother (W502) was purchased from Hunan Xinghui Washing Chemical Technology Development Co., Ltd. (Zhuzhou, China).

### 2.2. Preparation of Coal Gasification Fine Slag Residual Carbon Porous Material (GFSA)

KOH and GFSF samples were ground and mixed in a mortar at a predetermined ratio, which varied from 1 to 4 g/g, put into a porcelain boat, and heated in a tubular furnace for the activation test. The test was carried out under a continuous nitrogen flow of 200 mL/min and the heating rate was 10 °C/min. We heated the mixture to 750–900 °C and kept it at the final temperature for 30–120 min. After the activated sample was cooled to room temperature, it was rinsed to neutrality with distilled water, dried at 105 °C for 6 h, and sealed for storage.

### 2.3. Characterization Methods

N_2_ adsorption–desorption isotherms were obtained at −196 °C after degassing at 150 °C for 6 h using a volumetric sorption analyzer (IQ2MP-XR (Florida, USA)). The Brunauer–Emmett–Teller (BET) model was adopted for specific surface area analysis. The pore volume, pore size distribution, and average pore diameter were measured using the Barrett–Joyner–Halenda (BJH) and Horvath–Kawazoe (HK) models [31,32,33]. The morphologies of the samples were characterized using a field-emission scanning electron microscope (SEM, ZEISS Gemini 500 (Heidenheim, Germany)) with LnLen mode.

The cation exchange capacity (CEC) of the samples was determined according to a modified ammonium acetate method used in the literature [34]. The tests for each sample were implemented 3 times. The errors were less than ±5 mmol/100 g.

### 2.4. Adsorption Experiments

The adsorption experiment was conducted in a temperature-controlled water-bath shaker, with the GFSA sample obtained under the optimum condition. A stock MB solution (100 mg/L) was prepared by diluting MB with deionized water and experimental solutions were prepared from its dilution. The desired amount of the adsorbents was added to an MB solution (50 mL) with varying concentrations in a shake flask (100 mL). The pH of the solution was controlled by adding HCl (0.1 mol/L) and NaOH (0.1 mol/L). The shake flasks were shaken at 27 ± 0.5 °C with a stirring speed of 120 rpm for a certain time to achieve adsorption equilibrium. Then, the suspensions were filtered using membrane filters of 0.45 µm pore size. MB concentration in the solution was measured at 665 nm using a UV-2500 spectrophotometer (Shimadzu (Shanghai, China)). During this process, the effects of contact time, initial adsorbate concentration, adsorbent dosage, and pH on the adsorption process were investigated. The adsorption capacity was calculated by using Equation (1):(1)$$q_{e}=q_{t}=\left(C_{0}-C_{e,t}\right)V/m,$$
where *q_e_* is the adsorption capacity at equilibrium (mg/g), *q_t_* is the adsorption capacity at time t (mg/g), *C*_0_ is the initial concentration of MB (mg/L), *C_t_* is the MB concentration at time t (mg/L), *C_e_* is the MB equilibrium concentration (mg/L), *m* is the actual amount of GFAS (g), and *V* is the solution volume (L).

The removal efficiency (*η*) of the dye was determined using Equation (2):(2)$$\eta \left(\%\right)=100\left(C_{0}-C_{e}\right)/C_{0}$$

## 3. Results and Discussion

### 3.1. Effect of Activation Parameters on Preparation of GFSA

Cation exchange capacity (CEC) is an important index for rating adsorption materials, which reflects the cation exchange capacity of porous materials. The higher CEC of the adsorbent material means that there are a large number of adsorption sites inside the material, and its saturated adsorption capacity is also larger. The CEC of GFSF was 27.17 mmol/100 g. The influence of different operating conditions on the CEC value of GFSA is shown in Figure 2. In the experiment (Figure 2a), the activation temperature (800 °C) and the activation time (90 min) remained unchanged, GFSF/KOH mass ratios ranged from 1 to 4 g/g. The CEC of GFSA obtained by activation with different mass ratios firstly increased and then decreased. When the GFSF/KOH mass ratio was 2 g/g, the CEC of GFSA reached the maximum. The concise diagram of the reaction mechanism of KOH and C is shown in Figure 1. KOH reacted with C in GFSF (Equations (3)–(5)) to generate K_2_CO_3_, K_2_O, H_2,_ and CO in the activation process [35].
(3)$$4\mathrm{KOH}+C=K_{2}{\mathrm{CO}}_{3}+K_{2}O+2H_{2}↑$$
(4)$$K_{2}{\mathrm{CO}}_{3}+2C=2K+3\mathrm{CO}↑$$
(5)$$K_{2}O+C=2K+\mathrm{CO}↑$$

After the reaction rose to a certain temperature, the potassium ions (K) dispersed in GFSF reacted with carbon atoms (C) which led to the pores, wherein K_2_CO_3_ and K_2_O obtained by the reaction continued to react with C in GFSF to further form pores [16]. When the content of activator KOH was relatively low, KOH could only activate with a small amount of C in GFSF, and could not form enough pores. With the increase in KOH, more C in GFSF participated in the activation reaction, which gradually increased the number of micropores and mesopores. Therefore, the CEC of GFSA was increased. The selective activation of KOH consumed mainly the carbon atoms located at the active site and left a large number of pores in the carbon matrix so that the specific surface area and pore volume of the activated carbon increased [35]. However, as the mass ratio of GFSF/KOH increased, after the C in the active site of GFSF was completely reacted, the C on the pore framework participated in the reaction leading to the collapse of micropores or mesopores to form macropores, so the CEC value of GFSA decreased. Activation temperature also influenced the CEC of the activated sample, due to various reactions that occurred in the activation process. The CEC of GFSA was enhanced by the increase in activation temperature up to 800 °C, then decreased (Figure 2b) GFSF/KOH mass ratio (2 g/g) and activation time (90 min) remained unchanged). As the temperature of the reaction increased, the number of pores generated increased due to the acceleration of the activation reaction rate. Moreover, the boiling point of metallic potassium is 762 °C. When the temperature exceeded its boiling point, potassium vapor flooded into the generated pores and the interlayer of graphite microcrystals, which promoted the activation reaction and generated more pores [10,36]. Therefore, when the activation temperature was 750 °C to 800 °C, the adsorption sites increased because the material had more pores, and the CEC of GFSA increased significantly. At 850 °C and 900 °C, the activation reaction further intensified, and the surface area and pore volume declined dramatically, due to the merging and collapse of pores caused by the overreaction of carbon and intense release of gas [37]. Hence, the activation temperature of 800 °C was the best choice for subsequent experiments. The effect of activation time on the CEC of GFSA was studied at an activation temperature of 800 °C and KOH/CGS mass ratio of 2.0 g/g (Figure 2c). Activation times of 30 min and 60 min were not sufficient to generate a well-developed porosity, leading to a lower CEC of the activated samples. In the activation reaction process, there were a pore opening effect and a pore expanding effect [38]. In the early stage of the reaction process, a large number of micropores were generated, which was mainly the opening process of pores, so the CEC of activated samples in this process also increased. With the increase in reaction time, the effect of pore expansion was predominant, and the micropores and mesopores in the activated samples collapsed and merged to form macropores or the pores disappeared. Thus, when the activation time was extended by 120 min, the CEC of the activated samples was reduced again. According to the above, the optimal activation conditions for GFSA were as follows: the mass ratio of GFSF/KOH was 2 g/g, the activation temperature was 800 °C, and the activation time was 90 min. The CEC of GFSA was 110.68 mmol/100 g under the optimum conditions. Compared with the process conditions of biomass-based activated carbon studied by some researchers [39,40,41], the activation temperature of GFSF was higher and the time was longer. The reason may be that the production of GFS undergoes a high-temperature gasification and chilling process, in which the carbon that has not participated in the gasification reaction is carbonized at an excessively high temperature, so that the graphite crystallites in the carbonized product are changed in an orderly manner, and the gap between the crystallites is reduced. Therefore, the requirements for subsequent activation conditions are increased [42].

### 3.2. Characterization of Materials

The mineralogy and morphology of GFSA prepared under the optimal process conditions and GFSF were analyzed. Figure 3 shows the SEM images of samples with different CEC. GFSF is composed of irregular residual carbon particles and spherical mineral particles, and some spherical mineral particles have almost adhered to the surface of the residual carbon [31]. The sample with a CEC value of 27.17 mmol/100 g was GFSF, and its morphology is shown in Figure 3a. There were almost no pores on the surface of carbon particles and a small number of molten mineral particles on the surface. Due to the release of gases and volatile compounds, there were obvious pores on the external surface of the carbon matrix, as shown in Figure 3b (GFSF/KOH mass ratio: 1:2, activation temperature: 800 °C, activation time: 30 min) and 3c (GFSF/KOH mass ratio: 1:2, activation temperature: 750 °C, activation time: 90 min). However, due to the shorter activation time and lower activation temperature, the sample had fewer pores and was microporous [43,44]. The pore of Figure 3c was larger than that of Figure 3b, so the CEC of the sample in Figure 3c is slightly higher. In Figure 3d (GFSF/KOH mass ratio: 1:4, activation temperature: 800 °C, activation time: 90 min), more pore structures were observed, with slit pores appearing. Some block pores appeared on the surface of the carbon matrix shown in Figure 3f (GFSF/KOH mass ratio: 1:2, activation temperature: 800 °C, activation time: 120 min). This may be because of excessive activation time and too high mass ratio of GFSF/KOH, which caused the activation strength to be too high, which caused the pore size to become larger or collapse and reduce the CEC [35,37]. The CEC of the sample in Figure 3e (GFSF/KOH mass ratio: 1:2, activation temperature: 800 °C, activation time: 90 min) was the largest, with abundant and relatively uniform circular pores distributed on its surface.

N_2_ adsorption–desorption isotherms of GFSF and GFSA are shown in Figure 4, where GFSA was the sample prepared under the best process conditions. The GFSF and GFSA had a certain volume increment in all relative pressure ranges, which conformed to class IV adsorption isotherms and formed an obvious H4 hysteresis loop with the desorption curve (relative pressure range 0.4–0.99), indicating that they had a typical mesoporous structure [6,45]. As can be seen from Figure 4, the adsorption capacity of GFSA was significantly higher than that of GFSF. This is consistent with the SEM diagram in Figure 3. The GFSA shown in Figure 3e had complex pores, while the surface of the GFSF sample shown in Figure 3a had no obvious pores. The pore size distribution and pore properties of GFSF and GFSA are shown in Figure 5 and Table 1. The pores of GFSF mainly existed in the form of mesopores with fewer micropores, so its average pore size was larger than that of GFSA. The number of GFSA micropores obtained after GFSF activation was greatly increased (Figure 5), and the micropore volume (V_micro_) and micropore rate (V_micro_/V_total_) grew (Table 1), which was conducive to the application of adsorption. The total specific surface area (S_BET_) and total pore volume (V_total_) of GFSA were greater than those of the raw material GFSF.

### 3.3. Methylene Blue (MB) Adsorption Test

#### 3.3.1. Influence of Initial Solution pH

The adsorption performance of adsorbents under different initial solution pH was investigated (Figure 6). As the pH changed from 2 to 10, the removal efficiency of MB by GFSA increased from 87% to 98%. The main reason for this phenomenon is that MB is a cationic dye, most of which exists in the form of cations in aqueous solution, while a large number of H^+^ ions in low pH solution compete and occupy the adsorption sites of MB^+^, so the removal efficiency is low. With the increase in solution pH value, the adsorption potential energy of functional groups on the surface of GFSA was dehydrogenated, and the negative charge increased, and the competition of H^+^ ions in the solution was weakened, leading to the increase in adsorption capacity [46]. When the pH was greater than 8, the removal efficiency of MB by the adsorbent GFSA was above 97%, which achieved a good adsorption effect. Therefore, the pH of the initial solution in the following adsorption test is 8.

#### 3.3.2. Effect of Adsorbent Dosage

The adsorption performance of adsorbents under different dosages was investigated, and the results are shown in Figure 7. The removal efficiency of MB increased gradually and then tended to be stable with the increase in GFSA dosage. When the dosage of GFSA exceeded 3 g/L, the removal efficiency increased slowly and remained unchanged with the increase in adsorbent, indicating that the adsorption basically reached equilibrium when the dosage was 3 g/L. In addition, the equilibrium adsorption capacity (q_e_) of the adsorbent decreased with the increase in adsorbent dosage. When the dosage was less than 3 g/L, the descending speed was relatively slow, and the descending rate was significantly enhanced when the dosage was greater than 3 g/L. This may be because increasing the amount of adsorbent can provide more adsorption sites when the concentration of MB solution is fixed. Although more MB was adsorbed on the adsorbent, the utilization rate of adsorbent per unit mass decreased. Therefore, 3 g/L was chosen as the appropriate dosage.

#### 3.3.3. Adsorption Kinetics

The adsorption behavior of the activated sample GFSA at different contact times was investigated (Figure 8a). In the range of 0–270 min, the adsorption capacity of MB on GFSA increased rapidly with time, while the adsorption capacity of GFSA changed slowly and gradually stabilized with time from 270 to 360 min.

To better understand the adsorption kinetics of MB by GFSA, a pseudo-first-order kinetic model (PFO), pseudo-second-order kinetic model (PSO), and intraparticle diffusion model (IPD) were used to model the adsorption process [30,47], as shown in Figure 8b–d. The model parameters from the fitting calculation are shown in Table 2 and Table 3. The models are represented by
(6)$$\frac{1}{q_{t}}=\frac{1}{q_{e}k_{1}t}+\frac{1}{q_{e}}\;(\mathrm{PFO})$$
(7)$$\frac{t}{q_{t}}=\frac{1}{k_{2}q_{e}^{2}}+\frac{t}{q_{e}}\;(\mathrm{PSO})$$
(8)$$q_{t}=k_{d}\times t^{0.5}+C\;(\mathrm{IPD})$$
where *q_t_* (mg/g) and *q_e_* (mg/g) are the MB adsorption capacities at various and equilibrium times *t* (min), respectively; *k*_1_ (min^−1^), *k*_2_ (g·mg^−1^ min^−1^), and *k_d_* (g·mg^−1^ min^−0.5^) are the PFO, PSO, and IPD rate constants, respectively; and *C* is a constant that involves the thickness and the boundary layer.

As can be seen from the results in Table 2, since the correlation coefficient R^2^ of the PSO kinetic model is higher, the PSO kinetic model can more accurately describe the adsorption data than the PFO kinetic model. It indicates that there is an electron exchange between the adsorbent surface and adsorbate molecules [19]. In order to further clarify the control steps of the adsorption process rate, the mechanism of the adsorption process was described by the IPD model. As shown in Figure 8d, the plots of q_t_ against t^0.5 for MB adsorption by GFSA consisted of two linear parts, but the two straight lines did not pass through the origin of the coordinate. Therefore, it is inferred that intraparticle diffusion is not the only step of rate control, and many other adsorption mechanisms affect the rate at the same time. The correlation coefficient of the IPD model parameters (Table 3) R_1_^2^ (0.9724) was larger than R_2_^2^ (0.9363), and the adsorption rate constant K_1d_ (1.2848) was larger than K_2d_ (0.1001). These results indicate that the adsorption was mainly divided into two stages, and the outer surface of the adsorbent plays a major role. The first stage was the membrane diffusion process, MB molecules diffused from the solution to the outer surface of GFSA, and the adsorption rate of GFSA to MB was faster in the initial stage. In the second stage, MB molecules continued to diffuse into the microporous channels inside the GFSA and were adsorbed on the microporous surface, but the adsorption rate slowed down due to adsorption saturation.

#### 3.3.4. Adsorption Isotherms

Figure 9a shows the influence of different initial concentrations of MB on the adsorption process. The adsorption of MB onto the GFSA gradually increased as the MB concentration increased until a maximum value was achieved. Since the dosage and solution volume of GFSA was constant, the adsorption capacity increased with the increase in initial concentration, but the residual amount of adsorbate also increased, hence, the MB removal efficiency will decrease. When the initial concentration reached a certain value, the adsorption reached equilibrium, and the adsorption value did not continue to improve.

In this section, two nonlinear isotherm models of Langmuir isotherm and Freundlich isotherm are used to explain the distribution of adsorbate molecules in equilibrium [48]. These isotherms are represented by
(9)$$\frac{C_{e}}{q_{e}}=\frac{1}{K_{L}q_{m}}+\frac{C_{e}}{q_{m}}$$
(10)$$q_{e}=K_{F}C_{e}^{1/n}$$
where *C_e_* (mg/L) is the MB concentration at equilibrium, *q_e_* (mg/g) is the equilibrium adsorption capacity of MB, *q_m_* (mg/g) is the maximum adsorption capacity of MB, and *K_L_* (L/mg) and *K_F_* ((mg/g) (L/mg)^1/n^) are the constants of the Langmuir and Freundlich models, respectively.

The regression parameters of the Langmuir and Freundlich isotherms are shown in Table 4. The fitting determination coefficient R^2^ of the Langmuir isotherm was 0.9995, and the fitting determination coefficient R^2^ of the Freundlich isotherm was 0.7077. Therefore, in this concentration range, the Langmuir adsorption isotherm model was more suitable for the MB adsorption test. According to the Langmuir isotherm calculation, the maximum adsorption capacity of GFSA to MB was 18.78 mg/g, which was close to the experimental test value of 19.18 mg/g. It showed that the adsorption sites on the surface of GFSA were evenly distributed, and MB formed monolayer adsorption on the surface of GFSA, and after reaching equilibrium, no migration of adsorbate molecules through the surface of the adsorbent was observed [49]. The value of 1/*n* calculated by the Freundlich isotherm model was 0.16 in the range of 0–1, indicating that the adsorption of MB on GFSA was feasible. Under this condition, the adsorption process can proceed [50]. Compared with the adsorption performance of adsorbents prepared with different wastes as raw materials studied by other researchers (Table 5), the adsorption capacity of porous materials prepared with gasification fine slag as raw materials in this paper can also be accepted. The preparation of adsorption materials from gasification fine slag can be further studied.

## 4. Conclusions

In the present study, coal gasification fine slag residual carbon porous materials (GSFA) were synthesized by chemical activation with KOH from GFSF. The porous composite had the highest CEC (110.68 mmol/100 g) under the optimal operation condition of a GFSF/KOH mass ratio of 2 g/g, an activation temperature of 800 °C, and activation treatment time of 90 min. The GFSA with a total surface area of 574.02 m^2^/g was synthesized using GFSF. After activation, abundant pore structures were observed on the exterior surface of GFSA. The porosity of the porous material was the most important factor affecting its CEC. Furthermore, MB was selected as the target pollutant to evaluate the adsorption properties of the porous composite. The results show that the adsorption effect of GFSA on MB increased with the increase in the initial solution pH. The pseudo-second-order kinetic model was more suitable for the fitting of equilibrium data, indicating that chemical adsorption mainly controls adsorption. Using the intraparticle diffusion model to fit the adsorption process was mainly in the stages of fast membrane diffusion and slow pore diffusion. The adsorption isotherm used the Langmuir isotherm model, which showed that the maximum single-layer adsorption capacity was 18.78 mg/g. The results of this study demonstrated that solid waste GFS could be a reasonable raw material to produce low-cost porous adsorbent materials for the removal of MB.

## Figures and Tables

**Figure 1 molecules-26-06116-f001:**
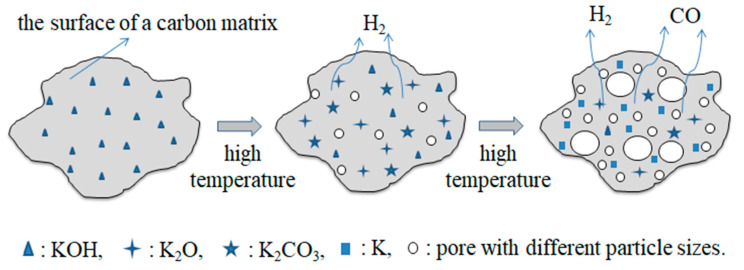
The concise diagram of the reaction mechanism of KOH and carbon in GFSF.

**Figure 2 molecules-26-06116-f002:**
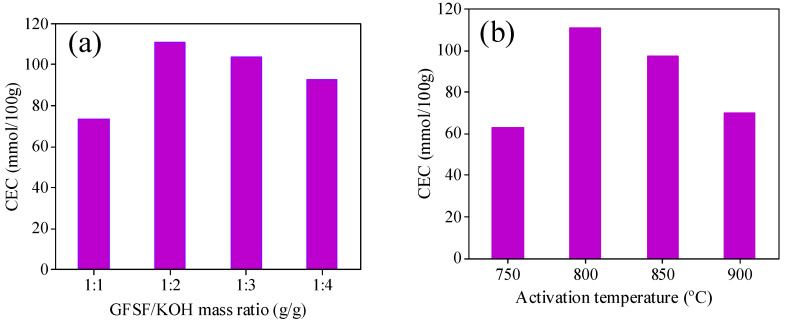
Effect of operation conditions on the cation exchange capacity (CEC) of GFSA: (**a**) gasification fine slag foam flotation obtained carbon residue (GFSF)/KOH mass ratio, (**b**) activation temperature, and (**c**) activation time.

**Figure 3 molecules-26-06116-f003:**
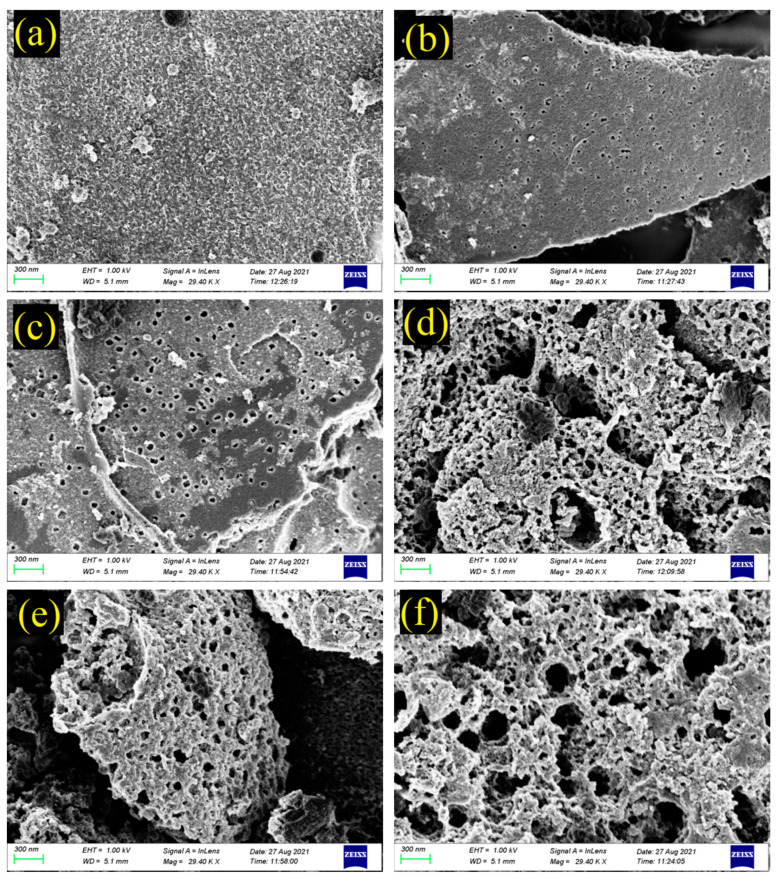
The SEM images of samples with different cation exchange capacity (CEC). (**a**) CEC = 27.17 mmol/100 g, (**b**) CEC = 48.57 mmol/100 g, (**c**) CEC = 62.85 mmol/100 g, (**d**) CEC = 92.51 mmol/100 g, (**e**) CEC = 110.68 mmol/100 g, and (**f**) CEC = 85.03 mmol/100 g.

**Figure 4 molecules-26-06116-f004:**
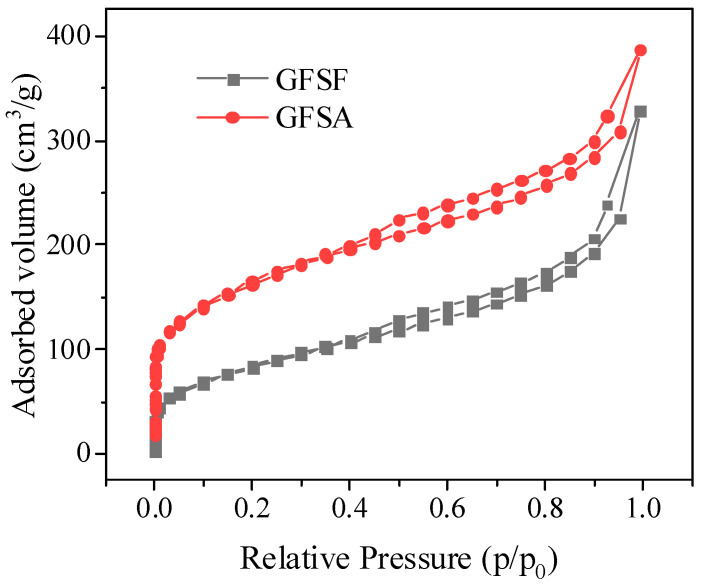
N_2_ adsorption–desorption isotherms of gasification fine slag foam flotation obtained carbon residue (GFSF) and coal gasification fine slag residual carbon porous material (GFSA).

**Figure 5 molecules-26-06116-f005:**
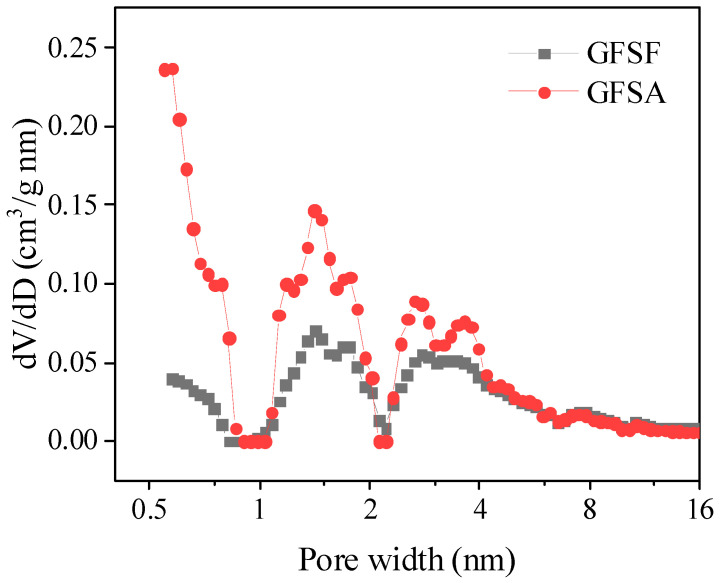
Pore size distribution of gasification fine slag foam flotation obtained carbon residue (GFSF) and coal gasification fine slag residual carbon porous material (GFSA).

**Figure 6 molecules-26-06116-f006:**
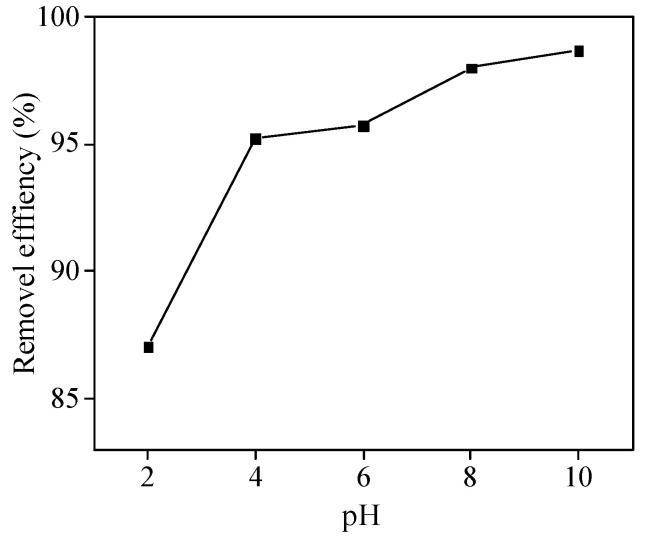
Influence of the solution pH on the removal efficiency of methylene blue (MB) by GFSA. (initial MB concentration: 50 mg/L, adsorbent dosage: 3 g/L, contact time: 360 min, and T = 27 ± 0.5 °C).

**Figure 7 molecules-26-06116-f007:**
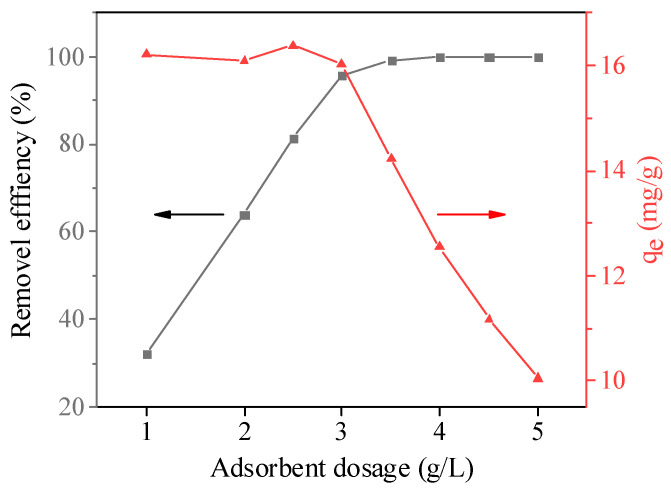
Influence of the adsorbent dosage on the removal efficiency of methylene blue (MB) and the MB adsorption capacity of the GFSA. (initial MB concentration: 50 mg/L, initial solution pH:8, contact time: 360 min, and T = 27 ± 0.5 °C).

**Figure 8 molecules-26-06116-f008:**
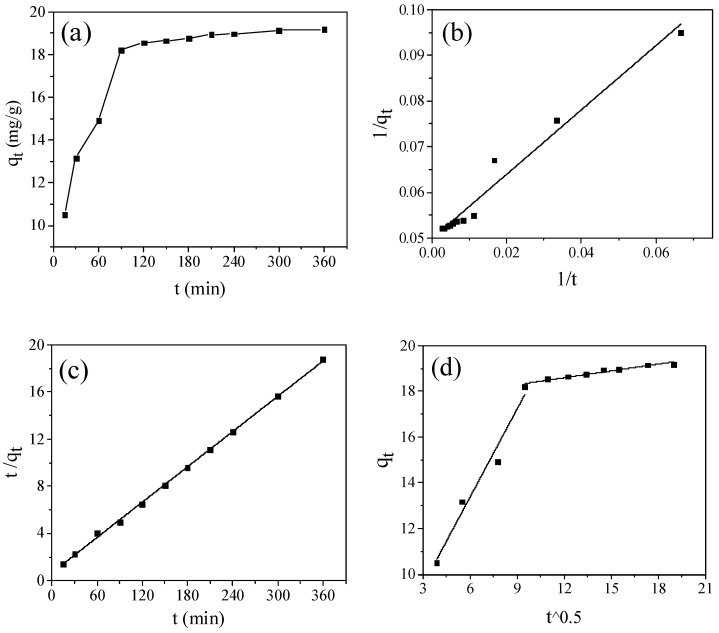
(**a**) Influence of time on the methylene blue (MB) adsorption capacity of the GFSA, (**b**) plots of 1/q_t_ vs. 1/t, (**c**) plots of t/q_t_ vs. t, and (**d**) plots of q_t_ vs. t^0.5^. (initial MB concentration:50 mg/L; adsorbent dosage: 3 g/L, initial solution pH: 8, and T = 27 ± 0.5 °C).

**Figure 9 molecules-26-06116-f009:**
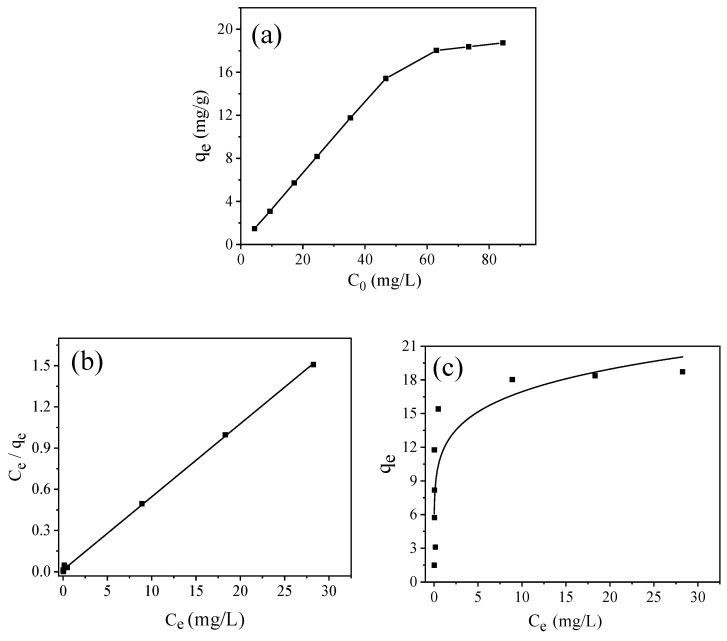
(**a**) Adsorption isotherm data for methylene blue (MB) adsorption onto the GFSA and (**b**) plots of C_e_/q_e_ vs. C_e_, (**c**) plots of q_e_ vs. C_e_. (adsorbent dosage: 3 g/L, contact time: 360 min, initial solution pH:8, and T = 27 ± 0.5 °C).

**Table 1 molecules-26-06116-t001:** Pore properties of gasification fine slag foam flotation obtained carbon residue (GFSF) and coal gasification fine slag residual carbon porous material (GFSA).

Samples	S_BET_ (m^2^/g)	V_total_ (cm^3^/g)	V_micro_ (cm^3^/g)	V_meso_ (cm^3^/g)	V_micro_/V_total_ (%)	V_meso_/V_total_ (%)	D_ave_ (nm)
GFSF	298.06	0.345	0.061	0.284	17.68	82.32	6.82
GFSA	574.02	0.467	0.166	0301	35.55	64.45	4.17

**Table 2 molecules-26-06116-t002:** Parameters of the pseudo-first-order (PFO) and pseudo-second-order (PSO) kinetic models.

Samples	q_e,exp_	PFO			PSO		
		k_1_	q_e,cal_	R^2^	k_2_	q_e,cal_	R^2^
GFSA	19.1803	0.0707	20.0545	0.9728	0.0037	20.0438	0.9992

**Table 3 molecules-26-06116-t003:** Parameters of the intraparticle diffusion (IPD) kinetic model.

Samples	Line 1: t = 0–90 min			Line 2: t > 120 min		
	k_1d_	C_1_	R_1_^2^	k_2d_	C_2_	R_2_^2^
GFSA	1.2848	5.6636	0.9724	0.1001	17.3940	0.9363

**Table 4 molecules-26-06116-t004:** Parameters of the Langmuir and Freundlich adsorption isotherms.

Samples	Langmuir Model			Freundlich Model		
	K_L_	q_m_	R^2^	K_F_	*n*	R^2^
GFSA	4.3449	18.7759	0.9995	11.6796	6.1828	0.7077

**Table 5 molecules-26-06116-t005:** Adsorption capacity of MB by adsorbents prepared from different waste materials.

The Raw Material of Adsorbent	Adsorption Capacity (mg/g)	Ref.
Date pits	17.3	[51]
Hazelnut shell	8.82	[52]
Kaolin	16.34	[53]
Coal fly ash	16.6	[54]
Cu_2_O-geopolymer	14.8	[55]
Wheat shells	16.56	[56]
GFSF	19.18	This research

## Data Availability

Not applicable.

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
