# Peer review of "Synthesis of Porous Material from Coal Gasification Fine Slag Residual Carbon and Its Application in Removal of Methylene Blue"

_molecules, 2021, doi:10.3390/molecules26206116_

Round 1

Reviewer 1 Report

This paper addresses an interesting approach for utilizing carbonaceous residues as adsorbents for a specific dye. This topic has been addressed abundantly by other groups, nevertheless, new advances and approaches are always welcome since the environmental issues are certainly critical, and practical solutions must be explored. I find this manuscript well written in general, with some grammar mistakes at specific sections (especially at the beginning of the manuscript). I would ask the authors to focus in the following points before publication on Molecules:

- The authors argue their method can be large-scaled at industrial levels, with simple operation and low cost. Is this real? since your described methods involve mortar ground and mixing, heating under nitrogen atmosphere, and rising with distilled water, at least.

- What it is 0.2 m3/h30 (page 3)?

- What the authors mean with "high-temperature chilling process" (page 5)?

- I suggest the scale bars in the SEM images (figure 3) should appear with bigger font size. They are quite difficult to distinguish in the current state.

- Section 3.3.4 lacks of deeper discussion on the 9c, and in my opinion, 9b can also be discussed further.

- Please check the grammar thoroughly for all the manuscript.

Reviewer 2 Report

This study is interesting. I have carefully read your manuscript and I think that the manuscript is generally well written. However, I have some minor issues that I need to discuss below to improve the work before the publication. Please, find my comments with attached file.

Reviewer 3 Report

line 192 - the typo: mesopores instead mesopods
Description of SEM images in lines 208 to 223 depicts the CEC values but is not related exactly to the preparation method. This is confusing and the reader may find it difficult to relate to the activation methodology for the investigated materials on this basis. You can add abbreviated references to activation parameters in parentheses.

The section 'Conclusions' contains observations rather than conclusions, hence I would call it a 'Summary'. Alternatively, the commentary on the obtained results should be extended. 

Reviewer 4 Report

The authors used a common waste, derived from coal gasification slag, for adsorption of pollutants from water. The subject is of high interest and the work is quite well described. I suggest the authors to make an effort to make it more clear to the readers, and to correct the following inaccuracies:

English errors in this sentence of the abstract should be corrected: “In this paper, Coal gasification fine slag residual carbon porous material (GFSA) was prepared by using gasification fine slag (GFS) foam flotation obtained carbon residue (GFSF) as raw material, and used as adsorbent to carry out adsorption test on waste liquid containing methylene blue (MB).”

Maybe “Coal” should be with lowercase letter. It is not clear what does the GFSF acronym stands for, it should be detailed letter by letter.

In particular, it is not clear the meaning of this part of the sentence: “Coal gasification fine slag residual carbon porous material (GFSA) was prepared by using gasification fine slag (GFS) foam flotation obtained carbon residue (GFSF) as raw material”.

Experimental section:

- the English language should be corrected, in particular changing the form from “instruction type” (open the air valve,  rinse with distilled water) to the passive form used everywhere else.

- “Until neutral” should be corrected in “until neutrality”

In the Results section, that expressed in the Figure 6 should be the removal efficiency, as explained in the experimental section, and not the removal rate.  Similar errors should be checked also in the text.

The results discussion clarity could be overall improved. 
